# Community social capital and the onset of functional disability among older adults in Japan: a multilevel longitudinal study using Japan Gerontological Evaluation Study (JAGES) data

Taiji Noguchi  ,[1,2] Katsunori Kondo,[2,3] Masashige Saito,[4] Hiroko Nakagawa-Senda,[1] Sadao Suzuki[1]

[1]Department of Public Health, Nagoya City University, Nagoya, Japan
[2]Center for Gerontology and Social Science, National Center for Geriatrics and Gerontology, Obu, Japan
[3]Center for Preventive Medical Science, Chiba University, Chiba, Japan
[4]Faculty of Social Welfare, Nihon Fukushi University, Chita-gun, Japan

**Correspondence to**
Dr Sadao Suzuki;
ssuzuki@med.nagoya-cu.ac.jp

## ABSTRACT

**Objective** The present study examined the association between community social capital and the onset of functional disability among older Japanese people by using validated indicators of social capital and a prospective multilevel design.

**Design** Prospective cohort study

**Setting** We used data from the Japan Gerontological Evaluation Study, established from August 2010 to January 2012 in 323 districts.

**Participants** The target population was restricted to non-institutionalised people aged 65 years or older who were independent in activities of daily living. Participants included 73 021 people (34 051 men and 38 970 women) who were followed up over a 3-year period.

**Primary outcome measure** The primary outcome measure was the onset of functional disability, defined as a new registration in public long-term care insurance system records with a care-needs level of two or above, analysed with multilevel Cox proportional hazards regression models by community social capital (civic participation, social cohesion and reciprocity).

**Results** The mean age of participants was 73.3 years (SD=6.0) for men and 73.8 years (SD=6.2) for women. During the study period, the onset of functional disability occurred in 1465 (4.3%) men and 1519 (3.9%) women. Of three community social capital variables, social cohesion significantly reduced the risk of onset of functional disability (HR 0.910; 95% CI 0.830 to 0.998) among men, after adjusting for individual social and behavioural variables. There was no significant effect among women.

**Conclusions** Living in a community with rich social cohesion is associated with a lower incidence of onset of functional disability among older Japanese men.

## INTRODUCTION

In almost every country, the proportion of older people is growing at an increasing rate,[1] and Japan has displayed the fastest growth. In 2012, 32% of the Japanese population was aged over 60 years, and this is expected to rise to 42% by 2050.[2] Age-related functional disability, defined as difficulty performing activities of daily living, is a very important public health issue worldwide.[3 4] Because functional disability affects health status and the costs of long-term care,[4] the prevention of functional disability among older people is increasingly important.

Recently, there have been great efforts to research the effect of social capital on health.[5–13] Putnam defines social capital as 'features of social organisation, such trust, norms and networks that can improve the efficiency of society by facilitating coordinated actions.'[5] There is considerable evidence of associations between social capital and various health indicators.[6–12]

## Strengths and limitations of this study

► This is the first prospective cohort study to examine the association between community social capital and the onset of functional disability among older people, by a large, nationwide population-based Japanese sample.

► To measure community social capital, an indicator consisting of validated multidimensional items was used, and we assessed three components of community social capital (civic participation, social cohesion and reciprocity).

► Multilevel survival analysis was used to examine community contextual characteristics for the onset of functional disability.

► More than 73 000 people aged 65 years or older participated and were followed up for 3 years period.

► While this study was a large sample size, the measurements were self-reported data.

Although both ecological and individual-level studies of social capital have yielded important insight, an appropriate examination of social capital as a collective (and contextual) influence on health requires multi-level analysis.[13] Prospective study designs are useful for establishing a valid relationship between social capital and health.[7] Several multilevel prospective studies have suggested contextual effects of social capital on health outcomes, mortality,[14 15] self-rated health,[16 17] suicide,[18] depression[19 20] and oral health.[21] Although two studies have reported an association between community social capital and the incidence of onset of functional disability among older people,[22 23] evidence remains insufficient. The study areas in this previous work were limited to certain parts of Japan, which limits the generalisation of the results. Additionally, these studies' measures of community social capital might not provide a full picture of social capital because the scales used might fail to capture the multiple dimensions of social capital, such as its cognitive and structural aspects.[24 25]

In the present study, we sought to examine the association between community social capital and the onset of functional disability among older people, using a prospective multilevel design and analysing data from a nationwide survey in Japan. We measured community social capital using an indicator consisting of recently developed and validated multidimensional items.[26]

## METHODS
### Study population
We used the Japan Gerontological Evaluation Study (JAGES) 2010–2013 cohort data.[27] Baseline data were collected using a self-administered questionnaire survey conducted from August 2010 to January 2012 among 85 161 people aged ≥65 years. The sample was restricted to people who did not already have functional disabilities, where functional disability was defined as being certified as eligible to receive long-term public care insurance (LTCI) system services. A simple random sample was obtained from the official residence registers in 13 large municipalities, and a complete census was taken of older residents residing in the remaining 11 smaller municipalities (response rate=66.3%).

Survey data from 81 980 respondents who provided information for identification by the public LTCI system were linked to the public LTCI records dataset over a 3-year follow-up period beginning 1 April 2010. We excluded 4549 respondents from 123 community areas with <50 respondents to avoid non-precise values from small sample sizes,[26] 253 respondents with unknown areas of residence, 1599 respondents who did not apply for public LTCI certification despite having basic activities of daily living (BADL) limitations, and to avoid the problem of reverse causation, 2558 respondents who did not complete the BADL items. Finally, we used data from 73 021 respondents in 323 community areas (figure 1).

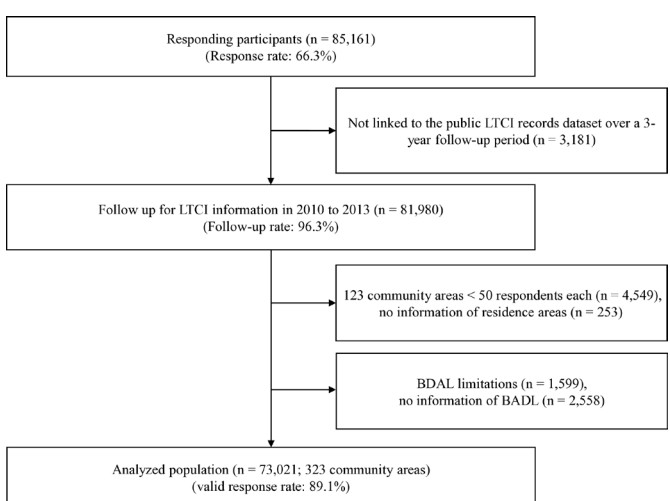

**Figure 1** Flow chart showing participation in the study cohort, 2010–2013. BDAL, basic activity of daily living; LTCI, long-term care insurance.

## Measurements
### Outcome
We collected information on the onset of functional disability from municipality-administered public LTCI records. The public LTCI system classifies frail older adults into seven levels ('support need levels' 1 and 2 and 'care need levels' 1–5; larger numbers indicate more severe need) using a nationally standardised and validated algorithm. This level is determined according to older adults' physical and mental care needs, regardless of informal care received,[28] and it is assessed both through computer-based and home-visit interviews with a trained healthcare professional and through examinations conducted by a primary physician.[29] In the computer-based assessment, time needed for care is calculated according to nine categories of care needs, including five BADL domains (bathing, eating, toileting, dressing and transferring), assistance with instrumental activities of daily living (IADL), behavioural problems, rehabilitation and medical services.[28] In our study, the onset of functional disability was defined as a new registration in the public LTCI records with a care-needs level of two or above, which requires at least 50 min of care daily and generally corresponds to needing any type of BADL care.[29] We used this outcome measurement because it has been found to reflect healthy life expectancy.[30]

### Community social capital
To measure community social capital, individual-level baseline data were aggregated for each of the 323 local districts. We assessed three components of community social capital (community civic participation, social cohesion and reciprocity), based on the instrument Saito *et al* developed and validated for measuring health-related community social capital.[26] Briefly, the community level was defined as the school district, a measure of community social capital was generated using factor analysis, and the factor scores for each small district were

used as community social capital variables.[22 26] Level of community civic participation was assessed by summing the percentages of participation in volunteer, sports and hobby groups in each community. Level of community social cohesion was measured by summing the percentages answering 'very' or 'moderately' (with possible response categories of 'very,' 'moderately,' 'neutral' 'disagree a little' and 'disagree') to three items: trust ('Do you think people living in your area can be trusted, in general?'), perception of others' intention to help ('Do you think people living in your area try to help others in most situations?') and attachment to the residential area ('How attached are you to the area where you live?'). Level of community reciprocity was measured by summing the percentages answering 'yes' to three items: receives emotional support ('Do you have someone who listens to your concerns and complaints?'), provides emotional support ('Do you listen to someone's concerns and complaints?') and receives instrumental support ('Do you have someone who looks after you when you are sick and confined to a bed for a few days?'). Community civic participation, social cohesion and reciprocity scores were standardised (subtracted from the mean and divided by the SD). We applied school districts as the community unit because this was the smallest feasible area unit identifiable in the JAGES data. School districts are likely to represent former 'villages', which existed before repeated municipality mergers took place in the last few decades in Japan. Civic activities are often conducted within each school district, and older people can easily travel on foot or by bicycle within the school district where they live.

### Individual responses on community social capital indicators

The individual-level social components used, which are closely related to the components of community social capital, were group participation in the community, perception of community social capital, social support and social isolation. Group participation in the community was measured as a count of participation in the following types of groups: volunteer, sports or hobby groups. Perception of community social cohesion was measured as a count of a study participant's responses of 'very' or 'moderately' to the following items: trust, perception of others' intention to help and attachment to the residential area. Social support was measured as a count of the number of the following types of social support each participant had: received emotional support, provided emotional support and received instrumental support. Social isolation was measured using the frequency of meeting with friends: A few times per year or less was considered moderate isolation, and more than once per month was considered non-isolation.

### Covariates

Sociodemographic characteristics and baseline health status were included in the analysis as covariates. These variables were age, equivalised income, educational attainment, marital status, self-rated health, self-reported body mass index (BMI), IADL, present illness, depression, lifestyle (smoking history, alcohol consumption and frequency of going outside) and individual social components. Age was categorised as 65–69, 70–74, 75–79, 80–84, or 85 years or older. Educational attainment was categorised as <6, 6–9, 10–12 or ≥13 years. Euivalised income was calculated by dividing the income of each household by the square root of the household size (number of family members); these figures were then categorised as low (<¥1 990 000; ¥120=US$1), middle (¥2 000 000–¥3 990 000) or high (≥¥4 000 000). We used this index as a measure of household economic status because it adjusts for household size. Marital status was categorised married, separated/divorced or never married. Living arrangement was categorised as living with others or living alone. Self-rated health was measured using a single question: 'What is your current health status?' with response options of 'excellent,' 'good,' 'fair' and 'poor.' BMI was categorised as <18.5, 18.5–24.9 or ≥25. IADL was assessed using a five-item subscale of the Tokyo Metropolitan Institute of Gerontology Higher Competence Scale.[31] We categorised those who had difficulty with at least one item as 'with difficulty'; others were categorised as 'without difficulty.' Present illness was measured using the following yes/no question: 'Do you receive treatment now?' Depression was assessed using the 15-item Japanese version of the Geriatric Depression Scale (GDS),[32] with scores categorised as no depression (0–4 points), depressive tendency (5–9 points) or depression (≥10 points). Smoking history was categorised as non-smoking, quit before 5 years, quit within 4 years or currently smoking. Alcohol consumption was categorised as never, past drinker or current drinker. Frequency of going outside was categorised as almost every day, one to three times per week, or once or twice per month or less. Urbanisation based on population density was categorised as urban (≥1500 people/km$^2$), suburban (1000–1500 people/km$^2$) or rural (<1000 people/km$^2$).

### Statistical analysis

The data included 73 021 individuals (first level) nested in 323 local districts (second level). The median number of subjects in each local district was 90 (25th and 75th percentile: 63 and 317). The multilevel analysis framework assumes that individual health outcomes are partly dependent on the districts where individuals live. Multilevel models estimate the variation in outcomes across districts (random effects) and the effects of community-level variables on the outcome, adjusting for individual compositional characteristics (fixed effects). Multilevel survival analysis using Cox proportional hazards regression models with stratification by sex was applied to calculate the HR and 95% CI for functional disability during the follow-up period. HRs were estimated for a 1 SD change in the percentages of community social capital variables. We used the following analysis models. First, the null model was used to assess whether the onset of functional disability varied across districts. Then, the

**Table 1** Respondent characteristics

| | Men (n=34051) | | | Women (n=38970) | | |
|---|---|---|---|---|---|---|
| | n | % | Incidence rate per 1000 person-years | n | % | Incidence rate per 1000 person-years |
| **Age (years)** | | | | | | |
| 65–69 | 11352 | 34.8 | 4.7 | 12018 | 32.1 | 2.7 |
| 70–74 | 9751 | 29.9 | 9.9 | 10993 | 29.4 | 5.6 |
| 75–79 | 7272 | 22.3 | 21.1 | 8570 | 22.9 | 14.7 |
| 80–84 | 4045 | 12.4 | 33.8 | 5006 | 13.4 | 34.7 |
| 85 or older | 1631 | 5.0 | 79.4 | 2383 | 6.4 | 79.1 |
| **Educational attainment** | | | | | | |
| <6 | 497 | 1.5 | 48.5 | 1211 | 3.1 | 59.4 |
| 6–9 | 14523 | 42.7 | 18.3 | 18903 | 48.5 | 14.0 |
| 10–12 | 10517 | 30.9 | 11.9 | 12272 | 31.5 | 10.4 |
| ≥13 | 6844 | 20.1 | 12.6 | 4332 | 11.1 | 9.7 |
| Other or missing | 1670 | 4.9 | 28.7 | 2252 | 5.8 | 25.6 |
| **Equivalised income** | | | | | | |
| Low | 22521 | 69.1 | 14.7 | 22708 | 60.6 | 12.6 |
| Middle | 5653 | 17.3 | 13.0 | 5495 | 14.7 | 11.5 |
| High | 925 | 2.8 | 10.3 | 832 | 2.2 | 13.5 |
| Missing | 4952 | 15.2 | 27.1 | 9935 | 26.5 | 20.1 |
| **Marital status** | | | | | | |
| Married | 28361 | 87.0 | 14.4 | 21903 | 58.5 | 8.6 |
| Separated/divorced | 3571 | 11.0 | 26.6 | 14132 | 37.7 | 22.0 |
| Never married | 444 | 1.4 | 9.8 | 815 | 2.2 | 20.9 |
| Other or missing | 1675 | 5.1 | 25.7 | 2120 | 5.7 | 21.8 |
| **Living arrangements** | | | | | | |
| Living with other | 30211 | 92.7 | 15.3 | 31306 | 83.6 | 13.6 |
| Living alone | 2287 | 7.0 | 19.1 | 6041 | 16.1 | 17.0 |
| Missing | 1553 | 4.8 | 27.5 | 1623 | 4.3 | 19.7 |
| **Body mass index (kg/m$^2$)** | | | | | | |
| <18.5 | 1798 | 5.5 | 38.9 | 3098 | 8.3 | 26.0 |
| 18.5–24.9 | 22990 | 70.6 | 14.1 | 24947 | 66.6 | 11.7 |
| ≥25.0 | 7199 | 22.1 | 11.5 | 7896 | 21.1 | 11.7 |
| Missing | 2064 | 6.3 | 36.4 | 3029 | 8.1 | 32.6 |
| **Self-rated health** | | | | | | |
| Excellent | 4208 | 12.4 | 6.8 | 4173 | 10.7 | 6.3 |
| Good | 22743 | 66.8 | 11.8 | 27015 | 69.3 | 11.3 |
| Fair | 5778 | 17.0 | 33.3 | 6351 | 16.3 | 27.7 |
| Poor | 1021 | 3.0 | 57.9 | 878 | 2.3 | 49.9 |
| Missing | 301 | 0.9 | 26.7 | 553 | 1.4 | 22.3 |
| **Present illness** | | | | | | |
| No | 8391 | 25.8 | 9.7 | 8454 | 22.6 | 9.5 |
| Yes | 23171 | 71.1 | 17.8 | 26981 | 72.0 | 15.5 |
| Missing | 2489 | 7.6 | 22.4 | 3535 | 9.4 | 17.9 |
| **Geriatric Depression Scale** | | | | | | |
| No depression | 21055 | 64.6 | 11.2 | 22164 | 59.2 | 10.1 |

Continued

| Table 1 | Continued | | | | | |
|---|---|---|---|---|---|---|
| | Men (n=34 051) | | | Women (n=38 970) | | |
| | n | % | Incidence rate per 1000 person-years | n | % | Incidence rate per 1000 person-years |
| Depressive tendency | 6063 | 18.6 | 22.3 | 6491 | 17.3 | 19.9 |
| Depression | 2013 | 6.2 | 29.3 | 2083 | 5.6 | 29.2 |
| Missing | 4920 | 15.1 | 24.7 | 8232 | 22.0 | 18.1 |
| Instrumental activities of daily living | | | | | | |
| Without difficulty | 22 452 | 68.9 | 10.8 | 31 079 | 83.0 | 7.8 |
| With difficulty | 8996 | 27.6 | 27.4 | 5188 | 13.9 | 49.5 |
| Missing | 2603 | 8.0 | 23.8 | 2703 | 7.2 | 25.2 |
| Alcohol consumption | | | | | | |
| Never | 18 187 | 53.4 | 11.6 | 5405 | 13.9 | 7.8 |
| Past | 2022 | 5.9 | 27.2 | 366 | 0.9 | 21.5 |
| Current | 11 710 | 34.4 | 20.5 | 30 709 | 78.8 | 15.1 |
| Missing | 2132 | 6.3 | 20.6 | 2490 | 6.4 | 18.4 |
| Smoking history | | | | | | |
| Non-smoking | 8191 | 25.1 | 14.4 | 31 089 | 83.0 | 13.6 |
| Non-smoking now, quit before 5 years | 13 967 | 42.9 | 16.0 | 1232 | 3.3 | 16.4 |
| Non-smoking now, quit within 4 years | 2934 | 9.0 | 16.6 | 446 | 1.2 | 12.8 |
| Smoking | 6305 | 19.3 | 16.5 | 1130 | 3.0 | 14.9 |
| Missing | 2654 | 8.1 | 20.8 | 5073 | 13.5 | 18.7 |
| Frequency of going outside | | | | | | |
| Once to twice a month or less | 1905 | 5.6 | 48.5 | 2758 | 7.1 | 40.8 |
| One to three times a week | 10 397 | 30.5 | 19.7 | 16 099 | 41.3 | 16.2 |
| Almost everyday | 19 632 | 57.7 | 10.6 | 17 799 | 45.7 | 8.1 |
| Missing | 2117 | 6.2 | 22.8 | 2314 | 5.9 | 20.1 |
| Social isolation | | | | | | |
| Non-isolation | 21 650 | 66.4 | 12.7 | 29 475 | 78.7 | 11.8 |
| Moderately isolation | 10 038 | 30.8 | 20.7 | 6261 | 16.7 | 21.4 |
| Missing | 2363 | 7.3 | 28.6 | 3234 | 8.6 | 25.1 |
| Group participation in the community | | | | | | |
| Non | 16 011 | 47.0 | 18.3 | 14 364 | 36.9 | 19.7 |
| One | 5562 | 16.3 | 9.3 | 6376 | 16.4 | 6.8 |
| Over two | 5180 | 15.2 | 7.9 | 6412 | 16.5 | 5.4 |
| Missing | 7298 | 21.4 | 22.8 | 11 818 | 30.3 | 17.1 |
| Social support | | | | | | |
| Non | 617 | 1.8 | 17.0 | 305 | 0.8 | 21.3 |
| One | 1260 | 3.7 | 17.8 | 690 | 1.8 | 21.3 |
| Over two | 30 004 | 88.1 | 15.3 | 35 523 | 91.2 | 13.4 |
| Missing | 2170 | 6.4 | 26.7 | 2452 | 6.3 | 25.5 |
| Perception of community social cohesion | | | | | | |
| Non | 3312 | 9.7 | 18.4 | 4345 | 11.1 | 14.3 |

Continued

**Table 1** Continued

| | Men (n=34051) | | | Women (n=38970) | | |
|---|---|---|---|---|---|---|
| | n | % | Incidence rate per 1000 person-years | n | % | Incidence rate per 1000 person-years |
| One | 5001 | 14.7 | 13.7 | 6205 | 15.9 | 12.1 |
| Over two | 23560 | 69.2 | 15.1 | 25592 | 65.7 | 14.1 |
| Missing | 2178 | 6.4 | 29.5 | 2828 | 7.3 | 22.1 |
| Urbanisation | | | | | | |
| Rural | 10594 | 32.5 | 18.0 | 13169 | 35.2 | 16.2 |
| Suburban | 18560 | 57.0 | 15.8 | 20278 | 54.1 | 14.0 |
| Urban | 4897 | 15.0 | 11.9 | 5523 | 14.7 | 10.3 |

effect of community social capital on the onset of functional disability was investigated, adjusting for age, educational attainment, equivalised income, marital status, living arrangements, BMI, self-rated health, present illness, IADL, alcohol consumption, smoking history and urbanisation (model 1). GDS score and frequency of going outside were then included (model 2). The final model also included social isolation, group participation in the community, social perception of community social capital and social support (model 3). Because there was frequently missing data on the covariates, a 'missing' category was created for the analyses. The significance level was set at p<0.05. We used R (V.3.4.3 for Windows) for all of the statistical analyses. Random effects models were estimated using the 'coxme' function (coxme package).[33]

### Ethical issues

JAGES respondents were informed that participation in the study was voluntary and that completing and returning the self-administered questionnaire by mail indicated their consent to participate in the study.

### Patient and public involvement

No patients were involved in the development of the research question, study design or data interpretation in this study.

### Results

Of the 73021 respondents over the follow-up period (average=2.7 years), 34051 were men and 38970 were women. The average age was 73.3 years (SD=6.0) for men and 73.8 years (SD=6.2) for women. During the follow-up period, 1465 (4.3%) new cases of functional disability occurred among men; among women, there were 1519 (3.9%) new cases of functional disability.

Table 1 shows the descriptive characteristics of the respondents at baseline. Respondents with onset of functional disability were more likely to be older, separated/divorced, presently ill, current or former drinkers, moderately socially isolated, and living in rural areas, and to have lower educational attainment, lower equivalised income, lower BMI, poor self-rated health, depression, IADL difficulties, lower frequency of going outside, no

community group participation and lower social support. These tendencies were almost identical for men and women.

Tables 2 and 3 show the results of multilevel the survival analyses for men and women, respectively. In the multivariable-adjusted model (Model 1), among men, a significant association was observed between community-level social capital and incidence of functional disability for 'social cohesion' (HR 0.904, 95% CI 0.824 to 0.992, p<0.05). This association was maintained after adding individual GDS score and frequency of going outside (HR 0.909, 95% CI 0.829 to 0.996, p<0.05; Model 2) and individual responses on community social capital indicators (HR 0.910, 95% CI 0.830 to 0.998; Model 3). Although the associations of incidence of functional disability with community-level civic participation and with reciprocity were not statistically significant, the point estimates for these effects were in the same direction, with HRs <1.0 (civic participation: HR 0.972, 95% CI 0.893 to 1.058; reciprocity: HR 0.920, 95% CI 0.829 to 1.021; Model 3). Among women, no significant association was observed (civic participation: HR 0.999, 95% CI 0.918 to 1.087; social cohesion: HR 0.930, 95% CI 0.847 to 1.020; reciprocity: HR 1.002, 95% CI 0.901 to 1.114; Model 3).

### DISCUSSION

To the best of our knowledge, this was the first study with a multilevel longitudinal design to examine the association between community social capital and the onset of functional disability using social capital indicators with verified validity in a large sample of older community-dwelling adults. The results suggested that living in a community with higher community social cohesion at baseline was associated with a lower future risk of functional disability, even after adjusting for individual responses on community social capital indicators. The present study indicated the importance of strategies to protect the health of older people through fostering cohesive communities with efforts such as promoting social connections and trust.

There are several possible pathways between community social cohesion and health. Social cohesion is

**Table 2** Multilevel survival analysis of functional disability among male respondents (n=34 051)

| | Model 1 | Model 2 | Model 3 |
|---|---|---|---|
| | HR (95% CI) | HR (95% CI) | HR (95% CI) |
| **Fixed effects** | | | |
| **Community-level variables** | | | |
| Community-level social capital | | | |
| Civic participation | 0.956 (0.878 to 1.041) | 0.959 (0.882 to 1.044) | 0.972 (0.893 to 1.058) |
| Social cohesion | 0.904 (0.824 to 0.992)* | 0.909 (0.829 to 0.996)* | 0.910 (0.830 to 0.998)* |
| Reciprocity | 0.930 (0.837 to 1.032) | 0.933 (0.841 to 1.035) | 0.920 (0.829 to 1.021) |
| Urbanisation (ref; urban) | | | |
| Suburban | 1.238 (0.977 to 1.568) | 1.230 (0.972 to 1.557) | 1.241 (0.981 to 1.571) |
| Rural | 1.278 (0.986 to 1.655) | 1.232 (0.951 to 1.596) | 1.251 (0.967 to 1.620) |
| **Individual-level variables** | | | |
| Age (ref; 65–69 years) | | | |
| 70–74 | 1.963 (1.603 to 2.403)** | 1.940 (1.584 to 2.376)** | 1.948 (1.591 to 2.387)** |
| 75–79 | 3.624 (2.989 to 4.394)** | 3.550 (2.928 to 4.305)** | 3.549 (2.924 to 4.307)** |
| 80–84 | 5.115 (4.189 to 6.246)** | 4.876 (3.990 to 5.958)** | 4.800 (3.923 to 5.873)** |
| 85 or older | 11.241 (9.126 to 13.846)** | 10.371 (8.407 to 12.795)** | 10.011 (8.099 to 12.375)** |
| Educational attainment (ref; <6) | | | |
| 6–9 | 0.717 (0.549 to 0.938)* | 0.738 (0.564 to 0.965)* | 0.738 (0.564 to 0.966)* |
| 10–12 | 0.682 (0.513 to 0.906)** | 0.720 (0.542 to 0.957)* | 0.727 (0.547 to 0.967)* |
| ≥13 | 0.745 (0.554 to 1.000) | 0.781 (0.581 to 1.050) | 0.793 (0.590 to 1.067) |
| Equivalised income (ref; low) | | | |
| Middle | 1.234 (1.043 to 1.459)* | 1.061 (0.903 to 1.247) | 1.067 (0.908 to 1.254) |
| High | 0.961 (0.523 to 1.769) | 1.015 (0.673 to 1.530) | 1.007 (0.668 to 1.519) |
| Marital status (ref; married) | | | |
| Separated/divorced | 1.028 (0.875 to 1.208) | 1.218 (1.029 to 1.441)* | 1.241 (1.048 to 1.470)* |
| Never married | 0.962 (0.638 to 1.449) | 0.907 (0.493 to 1.667) | 0.929 (0.504 to 1.713) |
| Living arrangements (ref; living with other) | | | |
| Living alone | 1.012 (0.799 to 1.282 | 1.013 (0.798 to 1.284) | 1.036 (0.815 to 1.317) |
| Body mass index, kg/m$^2$ (ref; 18.5–24.9) | | | |
| <18.5 | 1.688 (1.431 to 1.993)** | 1.651 (1.399 to 1.949)** | 1.633 (1.383 to 1.927)** |
| ≥25.0 | 0.919 (0.792 to 1.065) | 0.926 (0.798 to 1.074) | 0.932 (0.804 to 1.081) |
| Self-rated health (ref; excellent) | | | |
| Good | 1.356 (1.068 to 1.722)* | 1.289 (1.014 to 1.637)* | 1.265 (0.996 to 1.608) |
| Fair | 2.831 (2.205 to 3.636)** | 2.464 (1.910 to 3.178)** | 2.387 (1.850 to 3.079)** |
| Poor | 4.396 (3.287 to 5.881)** | 3.638 (2.699 to 4.903)** | 3.515 (2.606 to 4.740)** |
| Present illness (ref; no) | | | |
| Yes | 1.140 (0.978 to 1.327) | 1.143 (0.981 to 1.331) | 1.146 (0.983 to 1.334) |
| Instrumental activities of daily living (ref; without difficulty) | | | |
| With difficulty | 1.798 (1.606 to 2.013)** | 1.654 (1.475 to 1.856)** | 1.635 (1.457 to 1.835)** |
| Alcohol consumption (ref; non) | | | |
| Past | 1.089 (0.905 to 1.310) | 1.074 (0.893 to 1.292) | 1.055 (0.877 to 1.270) |
| Current | 0.850 (0.757 to 0.954)** | 0.866 (0.771 to 0.973)* | 0.873 (0.777 to 0.981)* |
| Smoking history (ref; non-smoking) | | | |

**Table 2** Continued

|  | Model 1 | Model 2 | Model 3 |
|---|---|---|---|
|  | HR (95% CI) | HR (95% CI) | HR (95% CI) |
| Non-smoking now, quit before 5 years | 1.089 (0.948 to 1.250) | 1.075 (0.936 to 1.234) | 1.071 (0.932 to 1.230) |
| Non-smoking now, quit within 4 years | 1.256 (1.021 to 1.545)* | 1.237 (1.006 to 1.522)* | 1.221 (0.993 to 1.502) |
| Smoking | 1.357 (1.153 to 1.598)** | 1.320 (1.120 to 1.554)** | 1.304 (1.107 to 1.536)** |
| Geriatric Depression Scale (ref; no depression) |  |  |  |
| Depression tendency |  | 1.224 (1.069 to 1.402)** | 1.214 (1.059 to 1.392)** |
| Depression |  | 1.222 (1.011 to 1.477)* | 1.236 (1.017 to 1.501)* |
| Frequency of going outside (ref; almost everyday) |  |  |  |
| One to three times a week |  | 1.237 (1.096 to 1.397)** | 1.225 (1.084 to 1.383)** |
| Once to twice a month or less |  | 1.899 (1.611 to 2.238)** | 1.801 (1.524 to 2.128)** |
| Social isolation (ref, non-isolation) |  |  |  |
| Moderately isolation |  |  | 1.060 (0.939 to 1.196) |
| Group participation in the community (ref; non) |  |  |  |
| One |  |  | 0.778 (0.645 to 0.937)** |
| Over two |  |  | 0.725 (0.588 to 0.894)** |
| Social support (ref; non) |  |  |  |
| One |  |  | 0.993 (0.625 to 1.578) |
| Over two |  |  | 1.287 (0.864 to 1.917) |
| Perception of community social cohesion (ref; non) |  |  |  |
| One |  |  | 0.852 (0.686 to 1.057) |
| Over two |  |  | 1.020 (0.853 to 1.220) |
| **Random effects** |  |  |  |
| Community-level variance | 0.0223 | 0.0210 | 0.0199 |

*P<0.05, **P<0.01.
Values presented are HRs and 95% CIs. Community-level social capital variables (civic participation, social cohesion and reciprocity) are 1 SD increase estimates. Variance of the intercept in the null model=0.0336.

determined by the resources available to members of tight-knit communities.[34] Cohesive communities might help residents to express trust towards their neighbours and to be psychologically healthier. Previous studies have revealed that neighbourhood social cohesion positively affected older people's subjective well-being,[35 36] and that cohesive communities prevented the occurrence of depressive symptoms in older people who lived alone and were at high risk of functional disability.[37] Thus, we considered high levels of community social cohesion to be potentially protective against the onset of functional disability via the positive effects on psychological health, such as enhancing subjective well-being and inhibiting depressive symptoms.

Two previous studies examined the association between community social capital and the onset of functional disability in older people using multilevel longitudinal designs.[23 24] Our results suggesting that higher community social cohesion was associated with lower risk of functional disability among men but not women were inconsistent with these previous studies. There are several possible reasons for this difference. First, both previous studies were surveys in a smaller area, compared with that in our research. Because our work used survey data from municipalities nationwide, the possibility for generalising our findings might be higher. Second, the measurement index of community social capital in the previous studies differed from ours. Both previous studies used only one item ('general trust') to measure community social cohesion. In contrast, we used multidimensional indicators consisting of three measurement items with verified validity, which might be more accurate for examining community contextual effects. Therefore, our results might reflect more accurate estimates of the effects of community social cohesion on individual health.

In the present study, community social cohesion affected the onset of functional disability only among men. It is likely that, among people who are currently in

**Table 3** Multilevel survival analysis of functional disability among female respondents (n=38 970)

| | Model 1 | Model 2 | Model 3 |
|---|---|---|---|
| | HR (95% CI) | HR (95% CI) | HR (95% CI) |
| **Fixed effects** | | | |
| **Community-level variables** | | | |
| Community-level social capital | | | |
| Civic participation | 0.986 (0.907 to 1.072) | 0.987 (0.908 to 1.072) | 0.999 (0.918 to 1.087) |
| Social cohesion | 0.936 (0.854 to 1.027) | 0.935 (0.853 to 1.025) | 0.930 (0.847 to 1.020) |
| Reciprocity | 1.003 (0.903 to 1.115) | 1.007 (0.906 to 1.119) | 1.002 (0.901 to 1.114) |
| Urbanisation (ref; urban) | | | |
| Suburban | 0.977 (0.772 to 1.236) | 0.968 (0.765 to 1.225) | 0.978 (0.772 to 1.238) |
| Rural | 1.045 (0.809 to 1.351) | 1.013 (0.784 to 1.309) | 1.024 (0.792 to 1.324) |
| **Individual-level variables** | | | |
| Age (ref; 65–69 years) | | | |
| 70–74 | 1.911 (1.478 to 2.469)** | 1.899 (1.469 to 2.455)** | 1.896 (1.466 to 2.451)** |
| 75–79 | 4.174 (3.293 to 5.290)** | 4.111 (3.242 to 5.213)** | 4.066 (3.203 to 5.161)** |
| 80–84 | 8.084 (6.371 to 10.257)** | 7.915 (6.232 to 10.052)** | 7.733 (6.079 to 9.838)** |
| 85 or older | 14.654 (11.434 to 18.780)** | 14.137 (11.016 to 18.143)** | 13.749 (10.694 to 17.677)** |
| Educational attainment (ref; <6) | | | |
| 6–9 | 0.723 (0.611 to 0.857)** | 0.729 (0.615 to 0.863)** | 0.734 (0.620 to 0.870)** |
| 10–12 | 0.679 (0.561 to 0.822)** | 0.687 (0.568 to 0.832)** | 0.701 (0.578 to 0.849)** |
| ≥13 | 0.792 (0.617 to 1.015) | 0.806 (0.629 to 1.034) | 0.826 (0.643 to 1.059) |
| Equivalised income (ref; Low) | | | |
| Middle | 1.037 (0.876 to 1.229) | 1.062 (0.896 to 1.259) | 1.063 (0.897 to 1.261) |
| High | 1.440 (0.989 to 2.097) | 1.505 (1.033 to 2.193)* | 1.500 (1.029 to 2.186)* |
| Marital status (ref; married) | | | |
| Separated/divorced | 1.078 (0.949 to 1.224) | 1.068 (0.940 to 1.213) | 1.064 (0.936 to 1.209) |
| Never married | 1.356 (0.983 to 1.870) | 1.330 (0.964 to 1.834) | 1.338 (0.969 to 1.848) |
| Living arrangements (ref; living with other) | | | |
| Living alone | 1.055 (0.912 to 1.221) | 1.044 (0.902 to 1.209) | 1.071 (0.924 to 1.241) |
| Body mass index, kg/m$^2$ (ref; 18.5–24.9) | | | |
| <18.5 | 1.423 (1.220 to 1.661)** | 1.395 (1.195 to 1.628)** | 1.363 (1.168 to 1.592)** |
| ≥25.0 | 0.997 (0.865 to 1.149) | 0.994 (0.862 to 1.146) | 1.003 (0.870 to 1.157) |
| Self-rated health (ref; excellent) | | | |
| Good | 1.404 (1.098 to 1.795)** | 1.350 (1.054 to 1.727)* | 1.333 (1.041 to 1.707)* |
| Fair | 2.354 (1.818 to 3.049)** | 2.111 (1.622 to 2.746)** | 2.078 (1.596 to 2.706)** |
| Poor | 3.439 (2.524 to 4.685)** | 2.934 (2.138 to 4.027)** | 2.856 (2.077 to 3.926)** |
| Present illness (ref; no) | | | |
| Yes | 0.943 (0.811 to 1.097) | 0.946 (0.813 to 1.101) | 0.954 (0.820 to 1.111) |
| Instrumental activities of daily living (ref; without difficulty) | | | |
| With difficulty | 2.449 (2.167 to 2.767)** | 2.287 (2.016 to 2.595)** | 2.171 (1.911 to 2.467) |
| Alcohol consumption (ref; non) | | | |
| Past | 1.574 (1.005 to 2.467)* | 1.550 (0.988 to 2.430) | 1.624 (1.035 to 2.548)* |
| Current | 0.915 (0.751 to 1.116) | 0.936 (0.767 to 1.141) | 0.949 (0.778 to 1.157) |
| Smoking history (ref; non-smoking) | | | |

Continued

**Table 3** Continued

| | Model 1 | Model 2 | Model 3 |
|---|---|---|---|
| | HR (95% CI) | HR (95% CI) | HR (95% CI) |
| Non-smoking now, quit before 5 years | 1.490 (1.121 to 1.981)** | 1.454 (1.093 to 1.934)* | 1.455 (1.094 to 1.936)* |
| Non-smoking now, quit within 4 years | 1.289 (0.758 to 2.193) | 1.291 (0.759 to 2.197) | 1.250 (0.735 to 2.129) |
| Smoking | 1.468 (1.077 to 2.000)* | 1.430 (1.048 to 1.949)* | 1.400 (1.026 to 1.911)* |
| Geriatric Depression Scale (ref; no depression) | | | |
| Depression tendency | | 1.232 (1.074 to 1.414)** | 1.240 (1.080 to 1.423)** |
| Depression | | 1.346 (1.116 to 1.624)** | 1.364 (1.127 to 1.651)** |
| Frequency of going outside (ref; almost everyday) | | | |
| One to three times a week | | 1.119 (0.985 to 1.272) | 1.098 (0.965 to 1.249) |
| Once to twice a month or less | | 1.323 (1.120 to 1.563)** | 1.257 (1.060 to 1.491)** |
| Social isolation (ref, non-isolation) | | | |
| Moderately isolation | | | 1.116 (0.977 to 1.275) |
| Group participation in the community (ref; non) | | | |
| One | | | 0.678 (0.555 to 0.829)** |
| Over two | | | 0.736 (0.587 to 0.923)** |
| Social support (ref; non) | | | |
| One | | | 0.727 (0.408 to 1.295) |
| Over two | | | 1.010 (0.619 to 1.647) |
| Perception of community social cohesion (ref; non) | | | |
| One | | | 0.912 (0.740 to 1.123) |
| Over two | | | 1.145 (0.961 to 1.364) |
| **Random effects** | | | |
| Community-level variance | 0.0202 | 0.0194 | 0.0209 |

*P<0.05, **P<0.01.
Values presented are HRs and 95% CIs. Community-level social capital variables (civic participation, social cohesion, and reciprocity) are 1 SD increase estimates. Variance of the intercept in the null model=0.0351

the older age groups, men sought and had stronger relationships with their colleagues before retirement than did women, given the nature of companies, particularly in Japan.[38] When this strong commitment is lost after retirement, these men may also experience a variety of changes in their living arrangements, leading to changes in their physical and mental health.[39] A cohesive community might be helpful in building new connections and encouraging social participation, which may keep men healthier and improve their psychological well-being. Honjo et al reported that rich social cohesion in a community buffered the risk of depression among older men living alone in Japan.[37] Thus, community cohesiveness may protect men's psychological health by helping them to build new connections in the community after retirement. However, further studies are needed to validate this hypothesis.

We considered urbanisation (population density), a potentially confounding characteristic of living areas. In exploratory analyses, we confirmed that urbanisation had a relatively strong influence as a confounding factor on the relationship between community social capital and the onset of functional disability. Therefore, the other characteristics of living area that were related to urbanisation, such as public security, might have caused residual confounding. However, we believe that this influence was relatively small because we adjusted for urbanisation as a representative factor of communities.

The present study had several strengths. First, using a large, nationwide population-based sample enabled us to conduct a community-level multilevel analysis to clarify the contextual relationship between community-level social capital and the onset of functional disability. Second, we used validated indicators consisting of multidimensional

items to measure community social capital. Therefore, we may have appropriately captured the whole of community social capital. However, the study also had several limitations. First, because the measurement was based on a self-administered questionnaire, the results are subject to response biases such as social desirability.[40] Social desirability bias may have artificially inflated social capital, which was calculated from the responses to the questionnaire. This, in turn, may have caused an overestimation of the relationship between community social capital and the onset of functional disability. Second, especially because the response rate to the survey was moderate (66.3%), selection bias might exist. Respondents in this study tended to be younger and healthier than the typical older adult population in the surveyed municipalities. In addition, people living in communities with low social capital might have been less likely than others to respond to the survey. These factors may have reduced the generalisability of our findings. However, because the respondents were randomly selected or completely enumerated from 24 municipalities in Japan, we believe that any effect of selection bias was small. Third, there were frequently missing data on the model variables. In the analyses, we dealt with these missing data using a 'missing' category.' This approach had the potential to bias the results. Therefore, we conducted sensitivity analyses by removing the missing data (complete case analyses). These analyses confirmed that the tendencies of the results were almost the identical when the missing data were removed (data not shown). Fourth, our study included no information about changes in social capital. Therefore, it is possible that unmeasured time-varying covariates such as economic changes or natural disasters may have biassed our results. Fifth, we used school district as the unit of analysis for communities because this was the smallest identifiable unit. However, the geographical scale of this unit may be slightly too large for the analysis of community-level social capital. Nevertheless, a school district represents an area of a size that older people can easily travel on foot or by bicycle, and community organisations, such as senior citizens' clubs and sports clubs, conduct their activities within individual school districts. Therefore, school district is a meaningful and appropriate unit of analysis for communities. Further work should build on our findings by defining regional units for spatial statistical analysis, using geographical information systems, for example. Finally, although our study was a prospective cohort study, the follow-up period was moderately short. Considering the possibility of reverse causation, study designs with a longer follow-up period are necessary in the future.

## CONCLUSION

In conclusion, this multilevel prospective cohort study found that higher levels of community social cohesion were associated with a lower incidence of onset of functional disability among older men, but not among older women, even after adjusting for individual social and behavioural variables. The findings suggest the importance of fostering cohesive communities to reduce the onset functional disability among older people.

**Acknowledgements** This study used data from the Japan Gerontological Evaluation Study (JAGES), conducted by the Center for Well-being and Society, Nihon Fukushi University, as one of their research projects. We thank Jennifer Barrett from Edanz Group (www.edanzediting.com/ac) for editing a draft of this manuscript.

**Contributors** TN had the idea for the study, participated in its design, performed the statistical analysis and drafted the manuscript as the principal author. KK, the principal investigator of the JAGES project, helped to develop the idea of the study, participated in acquiring the data and in designing the study, and critically revised the manuscript. MS helped to develop the idea of the study, participated in acquiring the data and in designing the study and critically revised the manuscript. HN-S helped with the data analysis and critically revised the manuscript. SS critically revised the manuscript. All authors read and approved the final manuscript.

**Funding** This study used data from the Japan Gerontological Evaluation Study (JAGES), which was supported by the Ministry of Education, Culture, Sports, Science and Technology of Japan's Supported Program for the Strategic Research Foundation at Private Universities (2009–2013), the Japan Society for the Promotion of Science KAKENHI (grant numbers JP18390200, JP22330172, JP22390400, JP23243070, JP23590786, JP23790710, JP24390469, JP24530698, JP24683018, JP25253052, JP25870573, JP25870881, JP26285138, JP26882010, JP15H01972), Health Labour Sciences Research Grants (H22-Choju-Shitei-008, H24-Junkanki (Seishu)-Ippan-007, H24-Chikyukibo-Ippan-009, H24-Choju-Wakate-009, H25-Kenki-Wakate-015, H25-Choju-Ippan-003, H26-Irryo-Shitei-003 (Fukkou), H26-Choju-Ippan-006, H27-Ninchisyou-Ippan-001), the Research and Development Grants for Longevity Science from Japan Agency for Medical Research and development (AMED) (JP18dk0110027, JP18ls0110002, JP18le0110009, JP19dk0110034), Research Funding for Longevity Sciences from the National Center for Geriatrics and Gerontology (24-17, 24-23, 29-42) and the World Health Organization Centre for Health Development (WHO Kobe Centre) (WHO APW 2017/713981). This work was supported by a grant from Nihon Fukushi University Research Institute for Health Sciences.

**Disclaimer** The views and opinions expressed in this article are those of the authors and do not necessarily reflect the official policy or position of the funding organisations.

**Competing interests** None declared.

**Patient consent for publication** Not required.

**Ethics approval** The ethics committee at Nihon Fukushi University approved the protocol and informed consent procedure for the present study (No. 10–05). This study conformed to the principles embodied in the Declaration of Helsinki.

**Provenance and peer review** Not commissioned; externally peer reviewed.

**Data availability statement** No data are available.

**ORCID iD**
Taiji Noguchi http://orcid.org/0000-0001-9165-5501

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
