## [Reviewer comments · BMJ Open]

ARTICLE DETAILS

TITLE (PROVISIONAL)	Community social capital and the onset of functional disability among older adults in Japan: A multilevel longitudinal study using Japan Gerontological Evaluation Study (JAGES) data
AUTHORS	Noguchi, Taiji; Kondo, Katsunori; Saito, Masashige; Nakagawa-Senda, Hiroko; Suzuki, Sadao

VERSION 1 - REVIEW

REVIEWER	Kathryn Morrison McGill University, Canada
REVIEW RETURNED	16-Apr-2019

GENERAL COMMENTS	This paper tackles an important subject in many western countries and the scope of the study is impressive with a very comprehensive dataset. The use of validated instruments strengthens the arguments. The writing is clear and the authors clearly know their field well. My primary concern relates to a lack of understanding of causal inference and how it relates to statistical methodology. For example, this sentence: "Moreover, to establish a causal relationship between social capital and health, observational studies must include prospective longitudinal analyses." Suggests that the authors are conflating design and analysis; lots of retrospective designs can be used legitimately in a causal framework. Given the explicit discussion early of the goal of causality, I wanted to see some discussion on confounding, measurement error, and selection bias explicit, and ideally some DAGS showing how they envision these relationships exist structurally. While there is a brief mention of bias in the discussion, there is no mention of how differential measurement error might occur, for example. Using a 'missing' category is generally considered a weaker option than multiple imputation. At a minimum, I'd like to see some sensitivity analyses here as compared to a complete case analysis.
---

	A more explicit spatial statistical approach may be appropriate here given the geographic nature of the multilevel units, and could have avoided excluding smaller (less populated) areas due to concerns about imprecision (e.g., lines 116-117). Minor issues: Typo in line 64: "be followed up"
--	---

REVIEWER	Ester Villalonga Olives University of Maryland School of Pharmacy USA
REVIEW RETURNED	06-May-2019

GENERAL COMMENTS	Thank you very much for inviting me to review the article entitled Community social capital and the onset of functional disability among older adults in Japan: A multilevel longitudinal study using Japan Gerontological Evaluation Study (JAGES) data. The JAGES is a great study that is making important contributions to the field. I have some suggestions to improve the paper:  1. It would be beneficial to have a consort diagram instead of a paragraph indicating how the study cohort was built. It would be clearer and only a brief explanation would be required. 2. I think there is some problem with the references. For example, line 147 in page 9 references the item 27, but I think it should be reference 26. Please check the entire document. 3. Could you please provide some rationale about the use of school district data? I understand it is the smallest area available but, is it a good unit of analysis? How similar is the population in a school area? 4. I am not familiar with the concept equivalized income. It would be great to have some explanation about this concept. 5. I am worried about the missing data of the study. How is the missing category affecting the results of the study? How were the missing treated? If the missing were not completely at random, how is this affecting the results? 6. The STOB Statement is very helpful. Is it possible to have some more explanation in the text about the checklist in the text? 7. It would be beneficial that an English speaker reviews the article.
--

VERSION 1 – AUTHOR RESPONSE

Reviewer 1

Comment #1

This paper tackles an important subject in many western countries and the scope of the study is impressive with a very comprehensive dataset. The use of validated instruments strengthens the arguments. The writing is clear and the authors clearly know their field well.

Response #1

Thank you very much for your comment. We believe that this is a very important report—not only for Japan, but also for all countries experiencing population aging. In our work, we aim to contribute the protection of health and quality of life for older people, with a focus on Japan, which has the highest proportion of older adults in the world.

Comment #2

My primary concern relates to a lack of understanding of causal inference and how it relates to statistical methodology. For example, this sentence:

"Moreover, to establish a causal relationship between social capital and health, observational studies must include prospective longitudinal analyses."

Suggests that the authors are conflating design and analysis; lots of retrospective designs can be used legitimately in a causal framework.

Response #2

Thank you for this suggestion. In the sentence you mention, what we intended to express was not in fact about causation, but rather about the validity of the relationship between community social capital and health. Furthermore, we intended to express that prospective study designs are one appropriate approach to achieving valid results. Upon reviewing this part of the text, we recognize that our statement was rather confusing, and we agree with your comment that this sentence needed revision. We have therefore changed this sentence as follows: "Prospective study designs are useful for establishing a valid relationship between social capital and health." (Lines 85–86)

In addition to this example, we found two other places in the manuscript that might suggest the conflation of study design and analysis, and we have revised both of these points in the text accordingly. First, under the "Objective" heading of the abstract, we have changed "prospective multilevel analysis design" to "prospective multilevel design." (Line 29) Second, we have changed the last sentence in the "Discussion" section from "Considering the possibility of reverse causation, analyses with a longer follow-up period are necessary in the future" to "Considering the possibility of reverse causation, study designs with a longer follow-up period are necessary in the future." (Lines 379–380)

Comment #3

Given the explicit discussion early of the goal of causality, I wanted to see some discussion on confounding, measurement error, and selection bias explicit, and ideally some DAGS showing how they envision these relationships exist structurally. While there is a brief mention of bias in the discussion, there is no mention of how differential measurement error might occur, for example.

Response #3

We agree with your assessment that the discussion on confounding, measurement error, and selection bias was insufficient.

To address the need for more discussion about potential confounding, we have made several revisions. Our analysis incorporated mainly individual socioeconomic status (SES; educational attainment and equivalized income) and population density of living areas (urbanization) as potential confounders because these factors are related to community social capital (civic participation, social cohesion, and reciprocity) and to the onset of functional disability (Subramanian S. et al., 2007; Wang R. et al., 2018). Variables gauging sociodemographic characteristics and baseline health status (e.g., age, marital status, and illness) were considered to be strongly associated with the outcome and were included in the analysis model as covariates. We considered depression, frequency of going outside, social isolation, and individual responses to social capital as intermediate factors. We confirmed that, other than age, these variables were not significant confounders. We show our assumed directed acyclic graph (DAG) to clarify our assumptions regarding the relationships among the variables (separate file; Image_figure A). We confirmed that the SES factors slightly confounded the relationship between community social capital and the onset of functional disability but did not reverse the results. In contrast, Urbanization was confirmed to be a significant negative confounder (data not shown). Therefore, characteristics of living area that are related to urbanization, such as public security, might have caused residual confounding. However, it is notable that, in previous studies, urbanization was adjusted using characteristics of living areas (Aida J. et al., 2012; Koyama S. et al., 2017; Nakagomi A. et al., 2019). Because urbanization is a representative characteristic of communities, the influence of residual confounding may have been relatively small. The following sentences were added to the “Discussion” section in the manuscript (Lines 336–343): “We considered urbanization (population density) a potentially confounding characteristic of living areas. In exploratory analyses, we confirmed that urbanization had a relatively strong influence as a confounding factor on the relationship between community social capital and the onset of functional disability. Therefore, the other characteristics of living area that were related to urbanization, such as public security, might have caused residual confounding. However, we believe that this influence was relatively small because we adjusted for urbanization as a representative factor of communities.”

We have also revised the manuscript to incorporate more discussion of measurement error. In our study, community social capital was assessed using a self-administered questionnaire, which may have resulted in biases such as social desirability. This may have caused measurement error and led to the artificial inflation of social capital as calculated from the responses to the questionnaire. This, in turn, may have led to an overestimation of the relationship between community social capital and the onset of functional disability. However, beyond acknowledging this possibility, we have no evidence indicating whether the measurement of community social capital used in the present study caused differential measurement error. Therefore, we cannot further discuss how this bias may have affected the findings. The following sentences were added to the “Discussion” section of the manuscript (Lines 350–355): “First, because the measurement was based on a self-administered questionnaire, the results are subject to response biases such as social desirability.[40] Social desirability bias may have artificially inflated social capital, which was calculated from the responses to the questionnaire. This, in turn, may have caused an overestimation of the relationship between community social capital and the onset of functional disability.”

Finally, we have revised the manuscript in response to your comment on selection bias. Although this survey covers multiple municipalities, the response rate was moderate (66.3%), which may indicate selection bias. Respondents in this study tended to be younger and healthier than the typical older adult population in Japan. In addition, people living in communities with low social capital might have been less likely than others to respond to the survey. This may have reduced the generalizability of our findings. However, because the respondents were randomly selected or completely enumerated from 24 municipalities in Japan, we believe that any effect of selection bias would be small. The following sentences were added to the “Discussion” section of the manuscript (Lines 355–362): “Second, especially because the response rate to the survey was moderate (66.3%), selection bias might exist. Respondents in this study tended to be younger and healthier than the typical older adult population in the surveyed municipalities. In addition, people living in communities with low social capital might have been less likely than others to respond to the survey. These factors may have reduced the generalizability of our findings. However, because the respondents were randomly selected or completely enumerated from 24 municipalities in Japan, we believe that any effect of selection bias was small.”

Comment #4

Using a 'missing' category is generally considered a weaker option than multiple imputation. At a minimum, I'd like to see some sensitivity analyses here as compared to a complete case analysis.

Response #4

Thank you for providing these insights. The statistical software package we used for the analyses could not perform multilevel Cox proportional hazards analysis after multiple imputation. Therefore, we dealt with missing data using a “missing” category in the analysis. However, this method of handling missing data may have biased the results. We conducted sensitivity analyses by removing the missing data (complete case analyses). These analyses confirmed that the tendencies of the results were almost the same (data not shown). The following sentences were added to the “Discussion” section of the manuscript (Lines 362–367): “Third, there were frequently missing data on the model variables. In the analyses, we dealt with these missing data using a “missing” category.” This approach had the potential to bias the results. Therefore, we conducted sensitivity analyses by removing the missing data (complete case analyses). These analyses confirmed that the tendencies of the results were almost the identical when the missing data were removed (data not shown).”

Comment #5

A more explicit spatial statistical approach may be appropriate here given the geographic nature of the multilevel units, and could have avoided excluding smaller (less populated) areas due to concerns about imprecision (e.g., lines 116-117).

Response #5

Thank you for your suggestion. As you point out, it would be more accurate to use a spatial statistical approach. Unfortunately, the dataset we used would not allow for this type of analysis. Therefore, we have added this topic to the discussion of the limitations of using school districts as multilevel units (Lines 370–378): “Fifth, we used school district as the unit of analysis for communities because this was the smallest identifiable unit. However, the geographic scale of this unit may be slightly too large for the analysis of community-level social capital. Nevertheless, a school district represents an area of a size that older people can easily travel on foot or by bicycle, and community organizations, such as senior citizens’ clubs and sports clubs, conduct their activities within individual school districts. Therefore, school district is a meaningful and appropriate unit of analysis for communities. Further work should build on our findings by defining regional units for spatial statistical analysis, using geographic information systems, for example.”

Comment #6

Minor issues:

Typo in line 64: "be followed up"

Response #6

Thank you for calling this to our attention. We have revised this phrase (Line 64).

Reviewer 2

Comment #1

Thank you very much for inviting me to review the article entitled Community social capital and the onset of functional disability among older adults in Japan: A multilevel longitudinal study using Japan Gerontological Evaluation Study (JAGES) data. The JAGES is a great study that is making important contributions to the field.

Response #1

Thank you very much for your comments. In our work, we aim to contribute the protection of health and quality of life for older people, with a focus on Japan, which has the highest proportion of older adults in the world.

Comment #2

It would be beneficial to have a consort diagram instead of a paragraph indicating how the study cohort was built. It would be clearer and only a brief explanation would be required.

Response #2

Thank you for this suggestion. We have created a flow chart of participants showing how the study cohort was built (figure 1). We have added a reference to this figure (separate file; Image_figure 1). However, we did not change the description of the cohort construction in the manuscript because we feel that the textual explanation is important for justifying our decisions in this process.

Comment #3

I think there is some problem with the references. For example, line 147 in page 9 references the item 27, but I think it should be reference 26. Please check the entire document.

Response #3

Thank you for this suggestion. We have revised this reference. In addition, we have checked the references throughout the manuscript to make sure that there were no additional errors.

Comment #4

Could you please provide some rationale about the use of school district data? I understand it is the smallest area available but, is it a good unit of analysis? How similar is the population in a school area?

Response #4

Here, you raise an important question. Although we used school district as the unit of analysis for communities, we do not have strong support for the validity of using this unit to evaluate community social capital. Historically, however, school districts are likely to represent the unit of former "villages," which existed before repeated municipality mergers took place in the last few decades in Japan. Government measures are implemented on a school district basis, children living in the same school district attend the same elementary school, and senior citizens' clubs also operate within school districts. Therefore, we believe that the school district is a suitable unit for evaluating community social capital. We have added a more detailed explanation of school districts to the manuscript (Lines 165–169): "School districts are likely to represent former 'villages,' which existed before repeated municipality mergers took place in the last few decades in Japan. Civic activities are often conducted within each school district, and older people can easily travel on foot or by bicycle within the school district where they live."

Comment #5

I am not familiar with the concept equivalized income. It would be great to have some explanation about this concept.

Response #5

You have raised an important question. Equivalized income was calculated by dividing the income of each household by the square root of the household size (family members). This indicator is used in many social epidemiological studies. We have added an explanation of equivalized income to the manuscript (Lines 192–195): Equivalized income was calculated by dividing the income of each household by the square root of the household size (number of family members); these figures were then categorized as low (< 1,990,000 JPY; 120 JPY = 1 USD), middle (2,000,000–3,990,000 JPY), or high (\geq 4,000,000 JPY). We used this index as a measure of household economic status because it adjusts for household size.

Comment #6

I am worried about the missing data of the study. How is the missing category affecting the results of the study? How were the missing treated? If the missing were not completely at random, how is this affecting the results?

Response #6

Thank you for raising these important questions. Because there were often missing values on the model variables, we dealt with the missing data using a “missing” category. Given the general tendency of people with low levels of social capital and those who are frail to avoid responding, the community social capital score may be underestimated. This may have caused an underestimation of the relationship between social capital and the onset of functional disability. We conducted sensitivity analyses by removing the missing data (complete case analyses). These analyses confirmed that the tendencies of the results were almost identical. In the revised manuscript, we have added the following sentences about the possible effects and limitations of the missing data (Lines 362–367): “Third, there were frequently missing data on the model variables. In the analyses, we dealt with these missing data using a “missing” category.” This approach had the potential to bias the results. Therefore, we conducted sensitivity analyses by removing the missing data (complete case analyses). These analyses confirmed that the tendencies of the results were almost the identical when the missing data were removed (data not shown).”

Comment #7

The STOBES Statement is very helpful. Is it possible to have some more explanation in the text about the checklist in the text?

Response #7

Thank you for this comment. We used the STOBES statement to check our report and made the necessary corrections. Major changes and additions to the text are shown below. In addition, we have attached a revised checklist based on the STOBES statement as part of this revised submission.

Item No.13 (c) Consider use of a flow diagram:

We have added a flow diagram (separate file; Image_figure 1).

Item No.19 Discuss limitations of the study, taking into account sources of potential bias or imprecision. Discuss both direction and magnitude of any potential bias:

We have added to our explanation of the study limitations in response to other comments from the reviewers (Lines 350–367): “First, because the measurement was based on a self-administered questionnaire, the results are subject to response biases such as social desirability.[40] Social desirability bias may have artificially inflated social capital, which was calculated from the responses to the questionnaire. This, in turn, may have caused an overestimation of the relationship between community social capital and the onset of functional disability. Second, especially because the response rate to the survey was moderate (66.3%), selection bias might exist. Respondents in this study tended to be younger and healthier than the typical older adult population in the surveyed municipalities. In addition, people living in communities with low social capital might have been less likely than others to respond to the survey. These factors may have reduced the generalizability of our findings. However, because the respondents were randomly selected or completely enumerated from 24 municipalities in Japan, we believe that any effect of selection bias was small. Third, there were frequently missing variables on the model variables. In the analyses, we dealt with these missing data using a “missing” category.” This approach had the potential to bias the results. Therefore, we conducted sensitivity analyses by removing the missing data (complete case analyses). These analyses confirmed that the tendencies of the results were almost the identical when the missing data were removed (data not shown).”

(Lines 370–378): “Fifth, we used school district as the unit of analysis for communities because this was the smallest identifiable unit. However, the geographic scale of this unit may be slightly too large for the analysis of community-level social capital. Nevertheless, a school district represents an area of a size that older people can easily travel on foot or by bicycle, and community organizations, such as senior citizens’ clubs and sports clubs, conduct their activities within individual school districts. Therefore, school district is a meaningful and appropriate unit of analysis for communities. Further work should build on our findings by defining regional units for spatial statistical analysis, using geographic information systems, for example.”

Item No.21 Discuss the generalisability (external validity) of the study results:

We have added some discussion of the generalizability of the study (Lines 355–362):

“Second, especially because the response rate to the survey was moderate (66.3%), selection bias might exist. Respondents in this study tended to be younger and healthier than the typical older adult population in the surveyed municipalities. In addition, people living in communities with low social capital might have been less likely than others to respond to the survey. These factors may have

reduced the generalizability of our findings. However, because the respondents were randomly selected or completely enumerated from 24 municipalities in Japan, we believe that any effect of selection bias was small.”

Comment #8

It would be beneficial that an English speaker reviews the article.

Response #8

Thank you for your suggestion. We asked a native English speaker to check this manuscript before submitting the revised version. We hope this will be sufficient to improve our use of English in the manuscript.

VERSION 2 – REVIEW

REVIEWER	Kathryn Morrison McGill University, Canada
REVIEW RETURNED	22-Jul-2019

GENERAL COMMENTS	I would like to thank the authors for such thorough work on their revision of this manuscript. I especially enjoyed the expanded discussions on confounding and measurement error. I think this paper is interesting and is suitable for publication in BMJ open.
---

REVIEWER	Ester Villalonga Olives University of Maryland School of Pharmacy, Baltimore (USA)
REVIEW RETURNED	07-Jul-2019

GENERAL COMMENTS	The reason why I have marked that this manuscript needs further statistical review is because the diagram that the authors say is a DAG is not a DAG per se. It is a causal figure of the relationships between the variables under study. The figure does not have the characteristics a DAG should have. A DAG flows in one direction to be acyclic. It is very important to review this part.
--